# Caregiver Willingness to Vaccinate Children with Pneumococcal Vaccines and to Pay in a Low-Resource Setting in China: A Cross-Sectional Study

**DOI:** 10.3390/vaccines10111897

**Published:** 2022-11-10

**Authors:** Linqiao Li, Yuan Ma, Wei Li, Guorong Tang, Yan Jiang, Huangcui Li, Shuxiang Jiang, Yun Zhou, Yuan Yang, Ting Zhang, Weizhong Yang, Libing Ma, Luzhao Feng

**Affiliations:** 1Department of Respiratory and Critical Care Medicine, Affiliated Hospital of Guilin Medical University, Guilin 541001, China; 2School of Population Medicine and Public Health, Chinese Academy of Medical Sciences & Peking Union Medical College, Beijing 100730, China; 3Center for Applied Statistics and School of Statistics, Renmin University of China, Beijing 100872, China; 4Guilin Center for Disease Control and Prevention, Guilin 541001, China

**Keywords:** peumococcal disease, pneumococcal conjugate vaccine, vaccine hesitancy, willingness to pay, low-resource setting

## Abstract

To determine the vaccine hesitancy of pneumococcal conjugate vaccines (PCVs) in a low-resource setting in China and to identify associated factors, a face-to-face questionnaire survey was conducted in the city of Guilin, China, from December 2021 to March 2022, which comprised sociodemographic information, attitudes toward vaccines and pneumonia, and PCV13 vaccination willingness and willingness to pay (WTP). Stepwise logistic regression and Tobit regression models were fitted to identify factors associated with PCV13 vaccination willingness and WTP, respectively. In total, 1254 questionnaires were included, of which 899, 254, and 101 participants showed acceptance, hesitancy, and refusal to vaccinate their children with PCV13, respectively. Only 39.07% of participants knew about PCV13 before this survey. A total of 558 (48.40%) participants accepted the full payment of vaccination, and 477 (41.37%) other participants accepted the partial payment, with a median cost of CNY 920.00. Demographics, social and psychological context, and attitudes toward vaccines were all associated with PCV13 vaccination but varied for hesitators and refusers. There is a substantial local demand for vaccinating children with PCV13 and partial payment is widely accepted. More publicity and educational efforts and a socially supportive environment are required to alleviate vaccine hesitancy.

## 1. Introduction

Pneumococcal disease (PD) poses a significant health threat to children worldwide. It is estimated that 380,900 deaths in children aged less than five years were attributed to PD in 2017 [1]. About 75% of cases of invasive pneumococcal disease (IPD) and 83% of cases of pneumococcal meningitis occur in children aged younger than two years [2]. The fatality rates of IPD could be up to 20% for septicemia and 50% for meningitis [2]. Furthermore, there is a risk of developing long-term neurological sequelae in survivors which can lead to a high economic burden. Pneumococcal infection often coincides with viral infections, such as influenza and respiratory syncytial virus, thus causing more serious clinical manifestations [2,3]. With the recent spread of the SARS-CoV-2 virus with the advent of COVID-19, the disease burden of pneumococcal infection could intensify.

Due to the high disease burden, the prevention of PD has been considered to be a priority by the World Health Organization (WHO) [4]; thus, the inclusion of pneumococcal conjugate vaccines (PCVs) in childhood immunization programs worldwide is recommended [2]. The PCVs have shown great efficacy in preventing pneumonia, vaccine-type IPD, and nasopharyngeal (NP) carriage [5], protecting the unvaccinated, and reducing antimicrobial resistance [2]. According to the Global Burden of Disease Study, increased PCV coverage led to a decrease of 6.3% in lower respiratory infection mortality among children aged under five years between 1990 and 2017 [6]. By the end of 2021, the pneumococcal vaccine had been introduced in 154 Member States [7]. However, it has not yet been included in the National Immunization Program (NIP) in mainland China. The three-dose PCV coverage rate was only 1.3% among children aged under five years in mainland China in 2017 [8], varying greatly by region and province. In economically developed cities, such as Shanghai, the initial (1st dose) and basic (3rd dose) immunization coverage in 2019 was 50.6% and 49.8%, respectively [9]. The vaccination coverage was even lower in other less developed regions. For example, for the city of Ningbo from 2017–2018, the initial immunization coverage was 11.3%; for the western regions in 2017, the average basic immunization coverage was estimated to be only 0.7% [8]. Thus, the current situation of low vaccination uptake needs to be improved.

Non-NIP vaccine, insufficient vaccine supply, a high vaccine cost, and age limits set on the administration of vaccines are possible reasons for low PCV vaccination coverage in China, which are expected to be addressed with the development of domestic vaccines [9]. However, vaccine hesitancy is a complicated issue that could be influenced by a broad range of factors. To help understand the underlying causes and address the challenge of vaccine hesitancy, many tools/measures have been developed [10]. For example, the vaccine hesitancy model “3C” was considered the most easily comprehensible, which comprises confidence, convenience, and complacency [11]. The “determinants of vaccine hesitancy matrix” model provides more detailed causative factors that can be divided into three categories: contextual, individual and group, and vaccine-specific issues [12]. The behavioral and social drivers (BeSD) tool was recently developed by the WHO, which pays more attention to social influences and practical issues along with thinking and feeling [13]. Integrated health behavior theories were also generated and implied to identify sociopsychological factors associated with vaccination uptake and to predict vaccination behavior [14]. However, vaccine hesitancy varies across time, place, and vaccines [11], thus making it difficult to fully explain the under-vaccination in a different context using existing tools or measures.

Considering the relatively high disease burden of PD and low PCV vaccination coverage among children in China, especially in low-resource settings (e.g., western regions of China) [8], developing efficient interventions to promote vaccination is pivotal. However, limited surveys on PCV hesitancy have been conducted in low-resource settings in China, and the intervention strategies needed there may differ from other settings, especially after the outbreak of COVID-19. Studies have indicated that factors arising from the low-resource context drove vaccine hesitancy, such as vaccine fear, mistrust, religion, myths, etc. [15]. Therefore, it is essential to identify the specific problems in the context of the Chinese low-resource setting and provide targeted solutions. Guilin, a less industrialized and multi-ethnic community in the western regions of China, is a representative low-resource setting with a less developed economy and limited healthcare and education resources [16]. Therefore, it was chosen for this study to investigate the perception of PCV13, willingness to vaccinate children with PCV13, and WTP for PCV13 vaccination among caregivers of children under five years old. The health behavior theoretical framework of Chu et al. [14] was utilized in the investigation of vaccine hesitancy to help understand the barriers to PCV13 vaccination in low-resource settings of China, and provide suggestions for local vaccination policy and intervention strategies accordingly.

## 2. Materials and Methods

### 2.1. Study Design and Data Collection

To understand the intention to vaccinate children with PCV13 and the WTP among parents in Guilin, China, a face-to-face questionnaire survey was designed and conducted by the School of Population Medicine and Public Health, the Chinese Academy of Medical Science and Peking Union Medical College, and Guilin Medical University. Initially, nine districts/counties were randomly selected from a total of 17 districts (district/county/city) in Guilin. Subsequently, all vaccination points (community health centers, maternity and childcare hospitals) from each sample district were selected to perform the survey. Face-to-face interviews in the vaccination points were carried out by trained interviewers of our research team with the help of staff from local vaccination points. From 15–23 December 2021, and 27–31 March 2022, all parents or grandparents who took their children aged under five years to get vaccinated at these vaccination points were invited to participate in the interview. We introduced our study to these parents and grandparents and asked them if they would like to participate in the survey. Those who agreed to participate and signed written consent informs were enrolled in our survey. All participants provided written informed consent forms before the interview. The accurate age of the child, the relationship between the participant and the child, and whether the participant has some decision-making power in the child’s vaccination were verified during the survey. The sample size was calculated using the following formula with α of 0.05 and a permissible error of 0.03: $n=\frac{z_{1-\alpha }^{2}\;\times \;p\;\times \;\left(1\;-\;p\right)}{d^{2}}$. According to previous studies, PCV13 coverage was assumed to be 50% [17]. Thus, for the current study, the estimated sample size was expanded by 10% to account for attrition. Therefore, the final sample size was 1174.

The inclusion criteria were as follows: the child had to be under five years old; the participant had to be a parent or grandparent of the child; the participant must have had some decision-making power in the child’s vaccination; there must have been no communication barriers. The exclusion criteria: the participant’s child was more than five years old; the participant was not the parent or grandparent of the child; the participant had no decision in the child’s vaccination; the participant did not sign written consent form; there were communication barriers.

The questionnaire comprised four parts: sociodemographic information (age, sex, educational attainment, annual household income, etc.), attitudes toward vaccines (importance, safety, effectiveness, preferred vaccine type, etc.), cognition about pneumonia and PCV13 (the harm of pneumonia to children, susceptibility to pneumonia for children, knowledge about PCV13, etc.), and PCV13 vaccination willingness and WTP (whether they accept vaccinating their children with PCV13, reasons that they refuse or hesitate about PCV13 vaccination, WTP for PCV13 vaccination, etc.). To make our analysis more logical and clear, the above factors were incorporated into the health behavior theoretical framework of Chu et al. [14]: Demographics; Social and psychological context (baseline vaccine hesitancy, cues to action, subjective norm); Risk perception of pneumonia; Attitudes and beliefs about PCV13; Intention of vaccination and WTP.

In terms of WTP, participants who refused to vaccinate their children with PCV13 were excluded. Four payment schemes for vaccination were considered sequentially: full payment, partial payment, service charge only, and completely free. If participants accepted one scheme, the remaining schemes were skipped. The Chinese monetary unit, yuan (CNY), was used. For the partial payment scheme, contingent valuation methods with a payment card approach were applied to reveal the maximum price participants were willing to pay for vaccination [18]. We gave participants values according to certain rules and asked them if they were willing to pay. The highest value was 2500, which is the full price for four doses of PCV13. The starting value was a random number ranging from 1300 to 1900. If it was accepted by participants, the next value would be the middle value between the starting value and 2500; otherwise, the next value would be the middle value between the starting value and 0. If participants still refused to pay after the process was repeated five times, they would be asked how much they would like to pay for vaccination in an open-ended question (ranging from CNY 1 to the last indicated value). The full vaccine price and service charge for 4-dose PCV13 were CNY 2500 and CNY 88, respectively. The full payment was calculated as the sum of the full vaccine price and service charge (CNY 2588). The partial payment for PCV13 vaccination is the sum of WTP and service charges. In our regression analysis, we excluded those who accepted none of the four payment schemes, although they did not show vaccine refusal.

For questions on attitudes toward vaccines and pneumonia, responses were measured on 3- or 5-point scales. To make our analysis clear, we categorized responses into two categories: “Yes” or “No.” For example, the original responses to the question “Do you think vaccines are important to children?” were “Important,” “Neutral,” and “Unimportant.” We considered the first group as “Yes” and the latter two groups as “No.”

### 2.2. Statistical Analysis

The participants’ demographic information, social and psychological context related to vaccination, pneumonia perception, and attitudes toward PCV13 were described in terms of the intention to get their children vaccinated with PCV13. Continuous variables were presented as medians (25th–75th percentile), and rank-sum tests were applied to make a group comparison, given that they were not normally distributed. Categorical variables were presented as counts (percentage), and chi-squared tests were applied to examine the differences among groups. A dumbbell chart was drawn to compare the reasons for vaccine refusal and vaccine hesitancy. In this plot, we categorized these reasons into “Thinking and feeling,” “Practical issues,” and “Others” according to the WHO BeSD of vaccination framework to make it more logical and comprehensible [13]. The waterfall plot and box plot were constructed to illustrate participants’ WTP, displaying the change in population proportions with different payment schemes and the accepted price for each scheme.

To identify associated factors for PCV13 vaccination willingness, stepwise binary logistic regression and multi-nomial logistic regression models were applied with dichotomous vaccination willingness (refusal or hesitancy versus acceptance) and trichotomous vaccination willingness (hesitancy versus acceptance, refusal versus acceptance) as the dependent variables. Candidate factors included participants’ demographics (age, sex, educational attainment, annual household income, etc.), social and psychological context (baseline vaccine hesitancy, whether the child has a history of pneumonia, whether they have been recommended PCV13 by health workers, etc.), risk perception of pneumonia (perceived pneumonia severity), attitudes toward PCV13 (perceived vaccine importance, whether PCV13 should be introduced to NIP, etc.). After variable selection, logistic regression models with these selected variables were rebuilt, and odds ratios (95% confidence intervals) were calculated.

The expenses that people were willing to pay were censored at CNY 0.00 (completely free) and CNY 2588.00 (full payment). To identify the factors associated with PCV13 payment willingness, a Tobit regression model with a forward selection method was applied, which has the advantage of estimating the relationship between explanatory variables and some censored dependent variables [19]. The dependent variable was expenses that participants were willing to pay, and the independent variables were participants’ sociodemographic information, attitudes toward vaccines and pneumonia, and the intention to get their children vaccinated. After the variable selection process, a Tobit regression model with these selected variables was rebuilt, and coefficient estimates and *p*-values were calculated. The WTP for each subgroup of participants was calculated and represented as medians (25th–75th percentile).

Data analyses were performed using SAS (version 9.4; SAS Institute Inc., Cary, NC, USA). The dumbbell chart, waterfall plot, and box plot were drawn by RStudio (version 2022.2.1.461; RStudio Team, 2022) with “ggplot2” and “waterfalls” packages. All statistical tests were two-sided, and a *p*-value of 0.05 indicated statistical significance.

### 2.3. Ethical Approval

The study protocol and questionnaire were approved by the Medical Ethics Committee of the Chinese Academy of Medical Sciences and Peking Union Medical College, Beijing, China (CAMS&PUMC-IEC-2021-035, 12 October 2021). All participants provided written informed consent forms before filling out the questionnaire.

## 3. Results

### 3.1. Demographic Characteristics of Participants

A total of 1409 questionnaires were returned. After excluding participants whose children were more than five years old (seven participants), those who were not the parents or grandparents of the child (five participants), and those who had no decision in the child’s vaccination (143 participants), 1254 participants remained in the analysis.

The sociodemographic characteristics of participants grouped by PCV13 vaccination willingness are shown in Table 1. The majority of participants were mothers, accounting for 75.76% of the total sample. Fathers and grandparents accounted for 22.65% and 1.59%, respectively. The median (interquartile range) ages of participants and children were 32.00 (29.00–35.00) years and 9.00 (4.00–19.00) months, respectively. Nearly 80% of participants had an annual household income of less than CNY 150,000. A total of 612 children were from only-child families, which accounted for 48.80%. A significant difference was noted among the different vaccination willingness groups in participants’ sex, educational attainment, relationship with the child, and whether the child was an only child. For example, the proportion of males in the vaccine hesitancy group (29.13%) was higher than that in both vaccine acceptance (22.25%) and refusal groups (13.86%). Participants who accepted vaccinating their children with PCV13 tended to have higher educational attainment and were from only-child families. We also found that the proportion of participants over 40 years old in the vaccine hesitancy group (15.35%) was larger than that in both vaccine acceptance (8.68%) and refusal groups (10.89%), though no statistical significance was found. The differences of vaccination intention among people with different characteristics are shown in Appendix A.

### 3.2. Behavioral and Social Factors Related to Vaccination

The behavioral and social factors related to vaccination are shown in Appendix A. In terms of social and psychological context, 1168 (93.14%), 896 (71.45%), and 894 (71.29%) participants held positive attitudes toward vaccines’ importance, safety, and effectiveness, respectively. A total of 190 children (15%) had a history of pneumonia. Additionally, 306 (24.40%) and 1025 (81.74%) participants had been recommended PCV13 by health workers and trusted health workers’ vaccination-related recommendations, respectively. Regarding risk perception of pneumonia, 92.34% of participants thought that pneumonia was harmful to their children, and these proportions in the acceptance group were about 20% higher than in the refusal group. For attitudes and beliefs about PCV13, only 39.07%, 24.00%, and 10.05% of participants knew the vaccine before the survey, the price of PCV13, and the diseases that PCV13 can prevent, respectively.

Publicity and educational efforts in communities or hospitals (43.88%) were the primary sources of PCV13-related knowledge, followed by the Internet (37.35%), friends and relatives (10.82%), and television, radio, and newspapers (7.14%) (Appendix A).

### 3.3. Willingness to Accept PCV13 Vaccination and Associated Factors

The study found that there were 899 (71.69%), 254 (20.26%), and 101 (8.05%) participants with vaccine acceptance, hesitancy, and refusal, respectively. According to the stepwise binary logistic regression (Table 2), demographics (only-child family), baseline vaccine hesitancy (perceived high vaccine effectiveness, acceptance of self-paid vaccines, and preference for imported vaccines), cues to action (history of pneumonia for the child), subject norms (the child’s siblings’ PCV13 vaccination history, trust in health workers’ vaccination recommendation), risk perception of pneumonia (perceived high severity of pneumonia), and attitudes and beliefs about PCV13 (more knowledge of PCV13, and the view that PCV13 should be introduced to NIP) were considered to be drivers of PCV13 vaccination. For example, participants who had an only child (OR: 0.48, 95% CI: 0.36–0.65), whose child had a history of pneumonia (OR: 0.63, 95% CI: 0.41–0.96), and whose older children had been vaccinated with PCV13 (OR: 0.13, 95% CI: 0.06–0.27) were less likely to refuse or hesitate to vaccinate children with PCV13.

Furthermore, we also compared hesitancy with acceptance, and refusal with acceptance, separately (Table 2). The associated factors were found to be slightly different from those in the binary logistic regression model. For example, those having been recommended PCV13 by health workers were less likely to have vaccine hesitancy (OR: 0.40; 95% CI: 0.19–0.82), and those knowing that PCV13 is not the vaccine for COVID-19 were more likely to have vaccine refusal (OR: 2.37; 95% CI: 1.29–4.36).

The reasons behind participants’ refusal and hesitancy about PCV13 vaccination were also surveyed and compared. As shown in Figure 1 and Appendix A, for participants with vaccine refusal, cost concerns (33.66%), insufficient vaccine knowledge (31.68%), and non-NIP vaccine (25.74%) were the top three reasons; while for those with vaccine hesitancy, insufficient vaccine knowledge was listed as the most important reason, accounting for 75.20%, followed by uncertainty about effectiveness (22.05%), cost concerns (22.05%), and uncertainty about safety (21.65%).

### 3.4. Willingness to Pay for PCV13 Vaccination and Associated Factors

The WTP was investigated among participants with no vaccine refusal. Participants’ responses to the different payment schemes are shown in the waterfall plot (Figure 2, Appendix A). A total of 70.51% of participants accepted the full payment scheme when the vaccine price was not informed. However, this percentage dropped 22.12% after they knew about the price. For the partial payment scheme, the percentage increased by 41.37%. For the service charge only and completely free schemes, the percentage increased by a further 7.89% and 1.04%, respectively. The expenses for each payment scheme are also shown (Figure 2). For example, the median value of partial payment was CNY 920.00.

A Tobit regression model with a forward selection method was used to investigate the factors associated with WTP. Participants who had an only child, who perceived high vaccine effectiveness, who were willing to vaccinate their child with self-paid vaccines, whose older children had been vaccinated with PCV13, whose family members had a medical background, who had knowledge about PCV13, and who accepted vaccinating children with PCV13 were willing to pay more for the vaccination. For example, participants who perceived high vaccine effectiveness were willing to pay CNY 500.86 more than those who did not. Additionally, participants with a preference for domestic vaccines and those who knew that PCV13 could not prevent COVID-19 tended to pay less. The median WTP values for each subgroup of these factors are shown in Table 3.

## 4. Discussion

The underlying causes of low PCV13 vaccination coverage are complicated and can be divided into two aspects: provider and demander. From the perspective of the provider, reliance on imported PCVs in China before the year 2020 caused a serious supply and demand gap [9], and the relatively narrow age limits of PCVs made it difficult for children to finish full vaccination if they missed the vaccination time. With the development of vaccine technology, two domestic PCV13 vaccines have been licensed for use, which is expected to solve the above-mentioned problems [17,20]. From the perspective of the demander, their low vaccination willingness might be the primary reason for under-vaccination, which could be influenced by many measurable and changeable behavioral and social factors [13]. The vaccination coverage in low-resource settings in China is much lower and vaccine hesitancy may have more specificity compared with other settings because of a less developed economy and limited healthcare services and education resources. Guilin is a typical low-resource setting in the western regions of China. We selected Guilin to investigate vaccination willingness and WTP among caregivers of children aged under five years, identified barriers to PCV vaccination from the perspective of the demander, and provided suggestions for local intervention strategies to increase PCV13 vaccination uptake. This would be valuable for other low-resource settings in China because common obstacles to PCV13 vaccination may exist.

In this study, only 39.07% of participants reported that they knew about PCV13 before this survey, and those who were clear about the diseases that PCV13 could prevent accounted for as little as 10.05%. This rate was much lower than another study conducted in three provinces in China in 2013, which showed that the knowledge rate about pneumonia was 52% [21]. To some extent, this could reflect insufficient local publicity and education efforts on PCV13. It is also noticeable that the knowledge level of PCV13 among participants with vaccine hesitancy was much lower than that in both vaccine acceptance and refusal groups; those who knew about PCV13 before were less likely to have vaccine hesitancy compared to vaccine acceptance. This is consistent with many other studies showing that limited knowledge about vaccines could be a barrier to vaccination uptake [11,22,23]. Therefore, publicity and education efforts on PCV13 should be promoted to reduce vaccine hesitancy, but it might have a limited impact on those with vaccine refusal because of the small difference in knowledge levels between the vaccine acceptance and refusal groups. 

The majority of participants (71.69%) were willing to vaccinate their children with PCV13, while 20.26% and 8.05% of participants showed vaccine hesitancy and refusal, respectively, which is similar to a survey conducted in the city of Weifang, Shandong Province, China [17]. An intention–action gap was observed for vaccination uptake [13,24], however, as per a prior study, intention for vaccination is a good predictor of actual behavior [25]. We explored the drivers for increasing vaccination intention, which can be adapted to promote vaccination behavior. As expected, high perceived vaccine effectiveness, high perceived severity of pneumonia, and health workers’ recommendations about vaccination were drivers of PCV13 vaccination, which has been indicated to be closely associated with vaccine hesitancy in many prior studies [11,17,26,27,28,29]. However, we did not find any association between annual household income and vaccination intention or WTP for PCV13, although the vaccine cost is a major concern for hesitancy. Furthermore, family characteristics influenced caregivers’ vaccination intentions. For example, only-child families or multi-child families in which older children have been vaccinated with PCV13 were more likely to accept it. For the one-child families, it could be explained by the fact that caregivers with an only child were much concerned about the child’s health, thus they might pay more attention to health knowledge and take a more positive attitude toward health behaviors, such as getting vaccinated. However, previous studies have reported opposite findings [30,31], which may be caused by different vaccines or settings. For multi-child families with older children vaccinated before, it could be more easily understood that they had already accepted the PCV13 vaccination or might display a positive attitude toward vaccination because of a positive peer effect [32].

It was noticeable that people who mistakenly believed that PCV13 was the vaccine for COVID-19 were less likely to refuse PCV13 vaccination and were willing to pay more for it. In other words, people showed greater willingness to vaccinate their children against COVID-19. This may be largely attributed to the social environment in which COVID-19 had become the focus of nationwide attention. For instance, vaccination campaigns have been advocated and vaccination recommendations are given not only from medical facilities but also from communities, workplaces, families, etc., which make people perceive the extremely high susceptibility and severity of COVID-19 [14,33]. It also indicates that a socially supportive environment can promote people to pursue a healthy life actively. Therefore, to increase awareness of the necessity and benefits of PCV13 vaccination, a more integrated and coherent strategy should be proposed to draw the social attention and cooperate with communities, families, and individuals, rather than relying only on healthcare workers. The corresponding interventions have already been identified by WHO BeSD tools and can be integrated into local intervention strategies, including community engagement, positive social norm messages (e.g., guidance and counseling of mass media), vaccine champions and advocates (e.g., financial support from the government), vaccination recommendations from health workers, etc. [13].

The top five reasons for participants with hesitancy or refusal were insufficient vaccine knowledge, cost concerns, uncertainty about the vaccine effectiveness and safety, and non-NIP of PCV13, all of which belong to the “Thinking and feeling” domain of the WHO BeSD tools, except for the cost concerns [13]. Promising interventions have also been developed for “Thinking and feeling” issues, such as campaigns to inform or educate the public about vaccination, and dialogue-based interventions [13]. However, the interventions for vaccine hesitancy and refusal might be different because the two groups tended to have different concerns about PCV13 vaccination, which was also found in another study about influenza vaccine hesitancy [31]. Participants with hesitancy had lower vaccine confidence, which suggested that more publicity and education efforts on PCV13 may be an effective way to turn vaccine hesitancy into acceptance. Meanwhile, for participants with vaccine refusal, the concerns were more involved with social context (non-NIP vaccine) and practical issues (high price), both of which might be solved through new vaccination policies with the development of domestic PCVs in the future [9]. Furthermore, participants with vaccine refusal had more complacency about children’s disease susceptibility and severity, which indicated that the education for these people should be focused on pneumococcal diseases.

The high vaccine cost is a practical issue hindering vaccination uptake. As indicated by a study conducted in Shanghai in China, the cost might make a difference in vaccine hesitancy between different non-NIP vaccines (Hib vaccine versus PCV) [34]. Significantly, the per capita gross domestic product (GDP) of Guilin in the year 2021 was CNY 46,767, which is much lower than the average level in China (CNY 80,976) [35,36]. The full payment of the PCV13 vaccination could be a financial burden for residents. When we asked about the acceptance of the full payment of the PCV13 vaccination, the proportion of participants who showed vaccine acceptance dropped by over 20% after we informed them of the cost. This finding demonstrates that paying the full price of PCV13 is a financial burden for a significant number of families [37,38,39]. When the price was completely free, the cumulative proportion reached 98.7%. In addition to financial reasons, the increasing acceptance could also be explained by the fact that government financial support for PCV13 may play a positive publicity role and reduce residents’ vaccine hesitancy [34]. It also reflected the limited economic affordability of residents, which should be considered when making local vaccination policies. Publicly funded PCV13 vaccination has been evaluated to be cost-effective and the inclusion of PCVs in the NIP is strongly recommended [40]. Furthermore, a one-free-dose policy for PCV13 has been adopted by the local government of Weifang, Shandong Province, and has been proved to be efficient to promote PCV vaccination [17].

This is the first study to investigate vaccine hesitancy regarding the administration of PCV13 in children and willingness to pay for it in low-resource settings in China. This study can help understand the current status of vaccine hesitancy that may differ from previous studies, because people may have updated their knowledge and attitudes about vaccination after the outbreak of COVID-19. Three different statuses of vaccine hesitancy (i.e., vaccine acceptance, hesitancy, and refusal) are considered, and the reasons and associated factors for vaccine hesitancy and vaccine refusal are investigated. This can help deal with vaccine hesitancy more efficiently because targeted interventions can be provided for different people. To evaluate willingness to pay among caregivers, four possible payment schemes for vaccination in China are considered (i.e., full payment, partial payment, service charge only, and completely free), and the detailed willingness to pay is further investigated using the contingent valuation methods with a payment card approach. It will help understand economic affordability of residents in depth and provide an important reference for local vaccination policy makers when they set the price for vaccines. Additionally, this study is a face-to-face interview survey, which could make our results more accurate and reliable.

This study had some limitations. First, a selection bias could exist because the participants were not selected through a random sampling method. However, all vaccination points from each sample district were selected, and all parents or grandparents who took children aged under five years to get vaccinated within a certain date range were invited to take part in the survey, which may make our participants more representative. Second, religion is reported to be associated with vaccination behavior; we only investigated religious concerns about vaccination but did not ask about the participants’ religious beliefs. The negative results regarding the association between religious concerns and willingness to vaccinate may thus be biased. Third, other confounders such as whether someone they knew got the disease, and vaccine availability, were not considered in this study.

## 5. Conclusions

With the widespread and continuous mutation of SARS-CoV-2, PCV13 vaccination is not only a preventive strategy for pneumococcal diseases but also a preparation for the local epidemic of COVID-19 to protect vulnerable populations because of the high risk of co-infection [2,3]. This study suggests that there is a substantial local demand for PCV13 vaccination among children under five years of age, but vaccine hesitancy still exists in this low-resource setting in China. Therefore, it is essential to promote PCV13 vaccination for children by ensuring an ample supply of vaccines and reducing parental concerns at the same time. People with vaccine hesitancy and refusal have different concerns, and targeted interventions should be applied. More publicity and education efforts on PCV13 may be an effective way to turn vaccine hesitancy into acceptance. New vaccination policies such as inclusion of PCVs in the NIP and publicly funded PCV13 vaccination are needed to address vaccine refusal. In addition, a socially supportive environment, such as community engagement and positive social norm messages, may promote people to get vaccinated actively. The full payment for PCV13 vaccination is a financial burden for a significant number of people, but the partial payment was widely accepted, indicating that the local government’s financial support and reasonable pricing should be appreciated. This study helps understand the barriers to PCV13 vaccination in low-resource settings of China, and provides suggestions for local vaccination policy and intervention strategies. Findings in this study could also be valuable for other regions, especially other low-resource settings in China where the PCV vaccination coverage is rather low and the same barriers for vaccination may exist.

## Figures and Tables

**Figure 1 vaccines-10-01897-f001:**
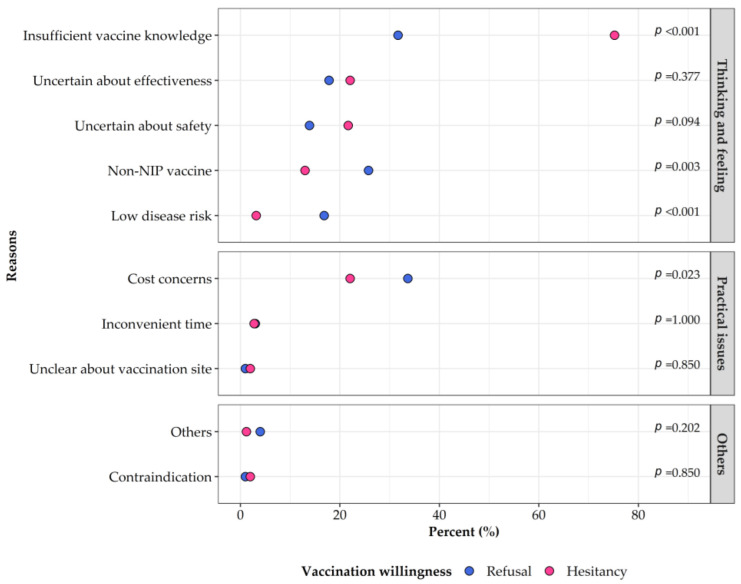
Reasons for refusal and hesitancy about getting children vaccinated with 13-valent pneumococcal conjugate vaccines. NIP is the abbreviation for National Immunization Program.

**Figure 2 vaccines-10-01897-f002:**
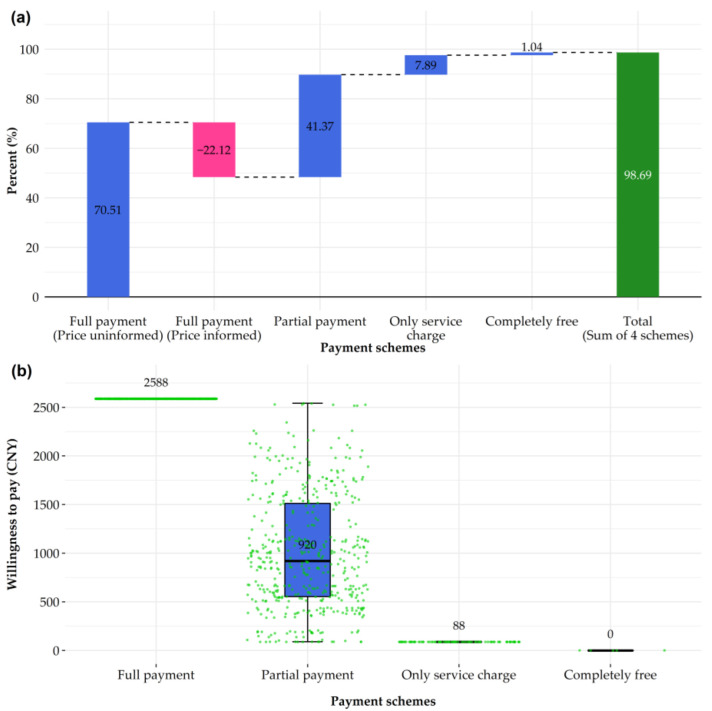
Willingness to pay for 13-valent pneumococcal conjugate vaccines excluding those with vaccine refusal. (**a**) Waterfall plot showing the change of participants’ proportions in different payment schemes. (**b**) Boxplot showing willingness to pay in different payment schemes.

**Table 1 vaccines-10-01897-t001:** Demographics of caregivers and their children.

	Total	Intention to Get the Child Vaccinated with 13-Valent Pneumococcal Conjugate Vaccines			*p*
		Acceptance	Hesitancy	Refusal	
*n* (%)	1254 (100.00)	899 (71.69)	254 (20.26)	101 (8.05)	
**Caregivers’ characteristics**					
Age (years), median (Q1-Q3)	32.00 (29.00–35.00)	32.00 (29.00–35.00)	32.50 (29.00–36.00)	33.00 (30.00–35.00)	0.158
Age group (years), *n* (%)					0.066
≤29	344 (27.43)	255 (28.36)	66 (25.98)	23 (22.77)	
30–34	492 (39.23)	352 (39.15)	99 (38.98)	41 (40.59)	
35–39	290 (23.13)	214 (23.80)	50 (19.69)	26 (25.74)	
≥40	128 (10.21)	78 (8.68)	39 (15.35)	11 (10.89)	
Sex, *n* (%)					0.005 *
Male	288 (22.97)	200 (22.25)	74 (29.13)	14 (13.86)	
Female	966 (77.03)	699 (77.75)	180 (70.87)	87 (86.14)	
Educational attainment, *n* (%)					<0.001 *
High school and below	488 (38.92)	314 (34.93)	127 (50.00)	47 (46.53)	
Bachelor’s degree and above	766 (61.08)	585 (65.07)	127 (50.00)	54 (53.47)	
Relationship to the child, *n* (%)					0.004 *
Mother	950 (75.76)	691 (76.86)	174 (68.50)	85 (84.16)	
Father	284 (22.65)	198 (22.02)	72 (28.35)	14 (13.86)	
Grandparent	20 (1.59)	10 (1.11)	8 (3.15)	2 (1.98)	
Annual household income (CNY 10,000), *n* (%)					0.055
<5	378 (30.14)	249 (27.70)	93 (36.61)	36 (35.64)	
5–15	597 (47.61)	433 (48.16)	116 (45.67)	48 (47.52)	
15–25	195 (15.55)	149 (16.57)	33 (12.99)	13 (12.87)	
≥25	84 (6.70)	68 (7.56)	12 (4.72)	4 (3.96)	
**Children’s characteristics**					
Sex, *n* (%)					0.351
Male	628 (50.12)	441 (49.11)	130 (51.18)	57 (56.44)	
Female	625 (49.88)	457 (50.89)	124 (48.82)	44 (43.56)	
Age (months), median (Q1-Q3)	9.00 (4.00–19.00)	9.00 (4.00–18.00)	9.00 (5.00–21.00)	11.00 (5.00–22.00)	0.123
Age group (months), *n* (%)					0.121
<6	393 (31.34)	292 (32.48)	70 (27.56)	31 (30.69)	
6–11	362 (28.87)	265 (29.48)	76 (29.92)	21 (20.79)	
12–23	271 (21.61)	193 (21.47)	51 (20.08)	27 (26.73)	
24–59	228 (18.18)	149 (16.57)	57 (22.44)	22 (21.78)	
An only child, *n* (%)	612 (48.80)	473 (52.61)	107 (42.13)	32 (31.68)	<0.001 *

* *p* < 0.05.

**Table 2 vaccines-10-01897-t002:** Factors associated with caregivers’ vaccine hesitancy or refusal of 13-valent pneumococcal conjugate vaccines.

	Refusal or Hesitancy versus Acceptance	Hesitancy versus Acceptance	Refusal versus Acceptance
**Demographics**			
Only-child family (Ref = “No”)	0.48 (0.36–0.65)	0.57 (0.41–0.79)	0.30 (0.19–0.50)
**Social and psychological context**			
**Baseline vaccine hesitancy**			
Perceived high effectiveness (Ref = “No”)	0.61 (0.44–0.84)	0.63 (0.44–0.90)	0.57 (0.35–0.94)
Willing to vaccinate their child with self-paid vaccines (Ref = “No”)	0.41 (0.26–0.64)	0.51 (0.31–0.83)	0.24 (0.13–0.44)
Vaccine preference (Ref = “No preference”)			
Domestic	1.09 (0.81–1.48)	-	-
Imported	0.50 (0.28–0.89)	-	-
**Cues to action**			
The child has a history of pneumonia (Ref = “No”)	0.63 (0.41–0.96)	-	-
**Subjective norm**			
The child’s siblings have been vaccinated with PCV13 (Ref = “No or unclear”)	0.13 (0.06–0.27)	0.11 (0.04–0.30)	0.15 (0.05–0.46)
Have been recommended PCV13 by health workers (Ref = “No”)	-	0.40 (0.19–0.82)	-
Trust health workers’ vaccination recommendation (Ref = “No”)	0.49 (0.34–0.70)	0.49 (0.33–0.73)	0.52 (0.30–0.90)
**Risk perception of pneumonia**			
Perceived high severity of pneumonia (Ref = “No”)	0.55 (0.33–0.92)	0.80 (0.45–1.43)	0.25 (0.13–0.48)
**Attitudes and beliefs about PCV13**			
Know that PCV13 is not the vaccine for COVID-19 (Ref = “No”)	-	-	2.37 (1.29–4.36)
Know PCV13 (Ref = “No”)	0.57 (0.41–0.78)	0.43 (0.25–0.73)	-
PCV13 should be introduced to NIP (Ref = “No or unclear”)	0.13 (0.09–0.20)	0.14 (0.09–0.22)	0.12 (0.07–0.22)

Note: Stepwise binary logistic regression and multi-nomial logistic regression models were applied with dichotomous vaccination willingness (Refusal or Hesitancy versus Acceptance) and trichotomous vaccination willingness (Hesitancy versus Acceptance, Refusal versus Acceptance), respectively. Candidate variables include: demographics (caregivers’ age, caregivers’ sex, relationship with the child, educational attainment, annual household income, child’s age, child’s sex, whether the child is an only child), social and psychological context (baseline vaccine hesitancy: perceived vaccine importance, perceived vaccine safety, perceived vaccine effectiveness, vaccine preference, willingness to vaccinate the child with self-paid vaccines; cues to action: whether the child has a history of pneumonia; subject norm: whether vaccination conflicts with religion, whether the child’s siblings have been vaccinated with PCV13, whether family members have a medical background, whether they have been recommended PCV13 by health workers, whether they trust health workers’ vaccination recommendations), risk perception of pneumonia (perceived pneumonia severity), attitudes and beliefs about PCV13 (whether they know PCV13, whether they know that PCV13 is not the vaccine for COVID-19, whether they know the price of PCV13, whether they know the diseases prevented by PCV13, whether PCV13 should be introduced to NIP).

**Table 3 vaccines-10-01897-t003:** Drivers for caregivers’ willingness to pay (WTP) to vaccinate children with 13-valent pneumococcal conjugate vaccines.

	Number of Participants, *n* (%)	WTP (CNY), Median (Q1-Q3)	Estimate	*p*
Total	1138 (100.0)	2260.00 (674.00–2588.00)	-	-
**Demographics**				
Only-child family				
Yes	575 (50.53)	2588.00 (968.00–2588.00)	634.16	<0.001
No	563 (49.47)	1542.00 (593.00–2588.00)	Ref	
**Social and psychological context**				
**Baseline vaccine hesitancy**				
Perceived high vaccine effectiveness				
Yes	832 (73.11)	2588.00 (903.50–2588.00)	500.86	<0.001
No	306 (26.89)	1138.50 (508.00–2588.00)	Ref	
Willing to vaccinate their child with self-paid vaccines				
No	92 (8.08)	383.00 (88.00–1022.00)	−1199.73	<0.001
Yes	1046 (91.92)	2588.00 (867.00–2588.00)	Ref	
Vaccine preference				
Domestic	523 (45.96)	1736.00 (613.00–2588.00)	−222.83	0.044
Imported	137 (12.04)	2588.00 (1058.00–2588.00)	285.89	0.107
No preference	478 (42.00)	2588.00 (886.00–2588.00)	Ref	
**Subject norm**				
The child’s siblings have been vaccinated with PCV13				
Yes	131 (11.51)	2588.00 (1060.00–2588.00)	590.06	0.001
No or unclear	1007 (88.49)	2005.00 (660.00–2588.00)	Ref	
Family members have a medical education background				
Yes	263 (23.11)	2588.00 (907.00–2588.00)	282.11	0.026
No	875 (76.89)	1853.00 (663.00–2588.00)	Ref	
**Attitudes and beliefs about PCV13**				
Know that PCV13 is not the vaccine for COVID-19				
No	406 (35.68)	2057.00 (613.00–2588.00)	264.18	0.042
Yes	732 (64.32)	2588.00 (842.00–2588.00)	Ref	
Know PCV13				
No	693 (60.90)	1640.00 (593.00–2588.00)	−434.90	0.001
Yes	445 (39.10)	2588.00 (1114.00–2588.00)	Ref	
**Motivation for vaccination**				
Intention to get the child vaccinated with PCV13				
Hesitancy	242 (21.27)	903.50 (346.00–2588.00)	−674.59	<0.001
Acceptance	896 (78.73)	2588.00 (971.00–2588.00)	Ref	

Note: Stepwise Tobit regression model was applied. Candidate variables include: demographics (caregivers’ age, caregivers’ sex, relationship with the child, educational attainment, annual household income, child’s age, child’s sex, whether the child is an only child), social and psychological context (baseline vaccine hesitancy: perceived vaccine importance, perceived vaccine safety, perceived vaccine effectiveness, vaccine preference, willingness to vaccinate the child with self-paid vaccines; cues to action: whether the child has a history of pneumonia; subject norm: whether vaccination conflicts with religion, whether the child’s siblings have been vaccinated with PCV13, whether family members have a medical education background, whether they have been recommended PCV13 by health workers, whether they trust health workers’ vaccination recommendations), risk perception of pneumonia (perceived pneumonia severity), attitudes and beliefs about PCV13 (whether they know PCV13, whether they know that PCV13 is not the vaccine for COVID-19, whether they know the price of PCV13, whether they know the diseases prevented by PCV13, whether PCV13 should be introduced to NIP), and motivation for vaccination (intention to get child vaccinated with PCV13).

## Data Availability

The datasets used and/or analyzed during the current study are available from the corresponding author on reasonable request.

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
