# Peer review of "Caregiver Willingness to Vaccinate Children with Pneumococcal Vaccines and to Pay in a Low-Resource Setting in China: A Cross-Sectional Study"

_vaccines, 2022, doi:10.3390/vaccines10111897_

Round 1

Reviewer 1 Report

This paper discusses the results of a survey related to the pneumococcal vaccination program for children. The results achieved can provide a statistical overview and implementation strategy for other countries. After the correction of text editing, I think I recommend accepting this manuscript to be published in Vaccines Journal.

Author Response

Thanks so much for your positive feedback.

Reviewer 2 Report

Thanks for the invitation to review this manuscript.

In the current study, Linqiao Li et al. investigated the attitude of the parents/guardians towards vaccination for pneumococcal disease. The study setting was Guilin, China, which is home for about 5 million residents. In addition, the authors aimed to investigate the willingness to pay for the vaccine. The importance of this study is related to the previous evidence that vaccine hesitancy is considered among the top global health threats and the huge burden of pneumococcal disease in children, which can be reduced by vaccination.

The study showed the gaps in knowledge pneumococcal disease and PCVs, with a considerable percentage (28%) of the participants who were hesitant towards PCV. The results can be helpful to inform local and global public health interventions regarding possible factors driving such phenomenon.

Overall, the study is well written with clear description of the valid methods. Importantly, the study limitations were mentioned by the authors. Therefore, I endorse this manuscript for publication following a minor revision for the following points:

1. In the Introduction, lines 36-37: “exclusively in children” might not be totally accurate since pneumococcal disease burden is known among adults especially those older than 65 years. Please revise accordingly.

2. In the Introduction, please cite a reference that support the statement that Guilin City is less industrialized with limited healthcare and education resources. This will be very important to justify the selection of the study setting.

3. In the Introduction, the authors can add a short statement on the utility of the Health Belief Model in the investigation of vaccination hesitancy since this framework was used in the study as indicated in Table 2 and Table S1.

4. In Table 1, to show the differences in a clearer way, I suggest using row percentages rather than column percentages.

5. A few suggestions to improve Figure 1 that can be considered by the authors: please use the font “Palatino Linotype” (this also applies for Figure 2), and remove the line connecting the two groups “hesitant” vs. “resistant”. Adding the p values to show the statistical significance between the two groups is also recommended.

Reviewer 3 Report

In this manuscript authors investigated the perception of PCV13, willingness to vaccinate their children with PCV13, and WTP for PCV13 vaccination among caregivers of children under five years old, to understand the barriers to PCV13 vaccination in low-resource settings of China and provided suggestions for local vaccination policy and intervention strategies accordingly.

The argument of the manuscript is original, but the manuscript has several methodological limitations. Inclusion and exclusion criteria must be clearly specified. Furthermore, discussion section lacks contents on the implications this study could have on national policies and what the paper adds to scientific literature. It is necessary that the authors investigate these aspects in depth, highlighting the areas in which these results could have healthcare implications. 

Especially results and discussion need a relevant revision both in stylistic terms and in contents. In general, the English in the paper can be understood.

The Major Essential Revisions include: 

-       In the materials and methods section authors should specify who carried out the data collection, how data collection with the questionnaire took place, how parents and grandparents were enrolled, describing these aspects in detail.

-       The inclusion criteria have been indicated in the materials and methods, but the exclusion criteria should also be made explicit. I suggest putting the inclusion and exclusion criteria in a separate paragraph.

-       Specify whether informed consent was given in written or oral form

-       Line 208: authors should specify the meaning of the values in brackets. Do the values indicate the interquartile range?

-       I suggest to report two decimal values in the percentages both in the text and in the tables.

-       Lines 213-214: I do not find this data in the table. In fact, I understand that in table 1 females are more hesitant to vaccinate than males (70.9%) and that the most hesitant age group was that between 30-34 years (39.0%). If I have misinterpreted these results, I ask for more explanation on the matter.

-       Furthermore, much of the data reported in Table 1 are not described in the text. I ask that they be included in the text.

-       Line 361-362: It would be important that the authors expand on this concept. In particular with regard to the recommendations given by health professionals who appear to be a driver for vaccination. This is a very interesting aspect, as not only for pneumococcal vaccination, but for vaccination itself, health professionals play a fundamental role in promoting vaccination and its acceptability, as they can influence the entire health behavior of their patients. However, suboptimal knowledge among healthcare professionals about the value of vaccination has been found to be one of the main obstacles to the success of vaccination campaigns. For this reason I suggest to deepen this topic by citing more scientific articles. For example: Pavia M. et al. Influenza and pneumococcal immunization in the elderly: Knowledge, attitudes, and practices among general practitioners in Italy. Public Health. 2003;117:202–207. doi: 10.1016/S0033-3506(03)00066-0.;Trucchi C. et al. Italian Health Care Workers' Knowledge, Attitudes, and Practices Regarding Human Papillomavirus Infection and Prevention. Int J Environ Res Public Health. 2020 Jul 22;17(15):5278. doi: 10.3390/ijerph17155278; Anastasi D. et al. Paediatricians knowledge, attitudes, and practices regarding immunizations for infants in Italy. BMC Public Health. 2009;9:463. doi: 10.1186/1471-2458-9-463.

-       Lines 415-419: It is important that the authors cite studies that also report the economic impact of the disease in terms especially of further hospitalizations and deaths to make people understand the importance of vaccination, for example: Ceyhan M. et al. Economic burden of pneumococcal infections in children under 5 years of age. Hum Vaccin Immunother. 2018;14(1):106–110. doi:https://doi.org/10.1080/21645515.2017.1371378; Amicizia D. et al. Economic burden of pneumococcal disease in children in Liguria, Italy. Hum Vaccin Immunother. 2022 Jun 13:2082205. doi: 10.1080/21645515.2022.2082205; Shiri T. et al. Pneumococcal disease: a systematic review of health utilities, resource use, costs, and economic evaluations of interventions. Value Health. 2019;22(11):1329–1344. doi:https://doi.org/10.1016/j.jval.2019.06.011

-       Authors should expand the conclusion section especially explaining what this study adds to the scientific literature. It is necessary that the authors investigate these aspects in depth, highlighting the areas in which these results could have healthcare implications.

-       Discuss in depth potential strengths of the study within a specific paragraph.

Author Response

Reviewer #3

Point 1: In the materials and methods section authors should specify who carried out the data collection, how data collection with the questionnaire took place, how parents and grandparents were enrolled, describing these aspects in detail.

Response 1: Thank you for your suggestions. Face-to-face interviews in the vaccination points were carried out by trained interviewers of our research team with the help of staff from local vaccination points. On December 15-23 2021, and March 27-31 2022, all parents or grandparents who took their children aged under five years to get vaccinated at these vaccination points were invited to participate in the interview. We introduced our study to these parents and grandparents and asked them if they would like to participate in the survey. Those who agreed to participate and signed written consent informs were enrolled in our survey. All participants provided written informed consent forms before the interview. The accurate age of the child, the relationship between the participant and the child, whether the participant has some decision-making power in the child’s vaccination were verified during the survey. The inclusion criteria were as follows: the child had to be under five years old; the participant had to be a parent or grandparent of the child; the participant must have had some decision-making power in the child’s vaccination; there must have been no communication barriers. The exclusion criteria: the participant’s child was more than five years old; the participant was not the parent or grandparent of the child; the participant had no decision in the child’s vaccination; the participant did not sign written consent form; there were communication barriers. We have added more details about the data collection in the Materials and Methods (page 3, line 108-118, 122-128).

Point 2: The inclusion criteria have been indicated in the materials and methods, but the exclusion criteria should also be made explicit. I suggest putting the inclusion and exclusion criteria in a separate paragraph.

Response 2: Thank you for your suggestion. The exclusion criteria were as follows: the participant’s child was more than five years old; the participant was not the parent or grandparent of the child; the participant had no decision in the child’s vaccination; the participant did not sign written consent form; there were communication barriers. We have added the exclusion criteria in in the Materials and Methods and put the inclusion and exclusion criteria in a separate paragraph (page 3, line 122-128).

Point 3: Specify whether informed consent was given in written or oral form

Response 3: Thank you for your suggestion. The informed consent was given in the written form. We have modified the sentence to make it more clearly. The sentence was modified as “All participants provided written informed consent forms before the interview” (page 3, line 115).

Point 4: Line 208: authors should specify the meaning of the values in brackets. Do the values indicate the interquartile range?

Response 4: Thank you for your suggestion. We agree that it will be more clearly by specifying the meaning of the values in brackets. It is true that the values indicate the interquartile range. We modified this sentence as “The median (interquartile range) age of participants and children were 32.00 (29.00-35.00) years and 9.00 (4.00-19.00) months, respectively.” (page 5, lines 219-220)

Point 5: I suggest to report two decimal values in the percentages both in the text and in the tables.

Response 5: Thank you for your suggestion. We modified our manuscript and supplementary file by using two decimal values in the percentages both in the text and in the tables.

Point 6: Lines 213-214: I do not find this data in the table. In fact, I understand that in table 1 females are more hesitant to vaccinate than males (70.9%) and that the most hesitant age group was that between 30-34 years (39.0%). If I have misinterpreted these results, I ask for more explanation on the matter.

Response 6: Thank you for your comments. We feel sorry that we did not describe it clearly. In table 1, females were more than males in all vaccine acceptance, hesitancy, and refusal groups, and that participants aged 30-34 years had the highest proportion in all vaccine acceptance, hesitancy, and refusal groups. Therefore, we cannot say that females were more hesitant to vaccinate than males, or participants aged 30-34 years were more hesitant to vaccinate. We modified the sentence in Lines 2113-214 as follows: “The proportion of males in vaccine hesitancy group (29.13%) was higher than that in both vaccine acceptance (22.25%) and refusal groups (13.86%).” “We also found that the proportion of participants over 40 years old in vaccine hesitancy group (15.35%) was larger than that in both vaccine acceptance (8.68%) and refusal groups (10.89%), though no statistical significance was found.” (page 5, lines 225-226, 228-231).

Point 7: Furthermore, much of the data reported in Table 1 are not described in the text. I ask that they be included in the text.

Response 7: Thank you for your suggestion. We have modified “3.1. Demographic characteristics of participants” and included more details in the text. The paragraph has been modified as follows (page 5, lines 216-231):

“The sociodemographic characteristics of participants grouped by PCV13 vaccination willingness are shown in Table 1. The majority of participants were mothers, accounting for 75.76% of the total sample. Fathers and grandparents accounted for 22.65% and 1.59%, respectively. The median (interquartile range) age of participants and children were 32.00 (29.00-35.00) years and 9.00 (4.00-19.00) months, respectively. Nearly 80% of participants had an annual household income of less than 150 thousand CNY. A total of 612 children were from only-child families, which accounted for 48.80%. A significant difference was noted among the different vaccination willingness groups in participants’ sex, educational attainment, relationship with the child, and whether the child was an only child. For example, the proportion of males in vaccine hesitancy group (29.13%) was higher than that in both vaccine acceptance (22.25%) and refusal groups (13.86%). Participants who accepted vaccinating their children with PCV13 tended to have higher educational attainment and were from only-child families. We also found that the proportion of participants over 40 years old in vaccine hesitancy group (15.35%) was larger than that in both vaccine acceptance (8.68%) and refusal groups (10.89%), though no statistical significance was found.”

Point 8: Line 361-362: It would be important that the authors expand on this concept. In particular with regard to the recommendations given by health professionals who appear to be a driver for vaccination. This is a very interesting aspect, as not only for pneumococcal vaccination, but for vaccination itself, health professionals play a fundamental role in promoting vaccination and its acceptability, as they can influence the entire health behavior of their patients. However, suboptimal knowledge among healthcare professionals about the value of vaccination has been found to be one of the main obstacles to the success of vaccination campaigns. For this reason I suggest to deepen this topic by citing more scientific articles. For example: Pavia M. et al. Influenza and pneumococcal immunization in the elderly: Knowledge, attitudes, and practices among general practitioners in Italy. Public Health. 2003;117:202–207. doi: 10.1016/S0033-3506(03)00066-0.;Trucchi C. et al. Italian Health Care Workers' Knowledge, Attitudes, and Practices Regarding Human Papillomavirus Infection and Prevention. Int J Environ Res Public Health. 2020 Jul 22;17(15):5278. doi: 10.3390/ijerph17155278; Anastasi D. et al. Paediatricians knowledge, attitudes, and practices regarding immunizations for infants in Italy. BMC Public Health. 2009;9:463. doi: 10.1186/1471-2458-9-463.

Response 8: Thank you for your valuable suggestion. We agree with you that health professionals play an essential role in improving the public’s vaccination uptake. We have added these citations correspondingly (page 13, line 386, references 26-29).

Point 9: Lines 415-419: It is important that the authors cite studies that also report the economic impact of the disease in terms especially of further hospitalizations and deaths to make people understand the importance of vaccination, for example: Ceyhan M. et al. Economic burden of pneumococcal infections in children under 5 years of age. Hum Vaccin Immunother. 2018;14(1):106–110. doi:https://doi.org/10.1080/21645515.2017.1371378; Amicizia D. et al. Economic burden of pneumococcal disease in children in Liguria, Italy. Hum Vaccin Immunother. 2022 Jun 13:2082205. doi: 10.1080/21645515.2022.2082205; Shiri T. et al. Pneumococcal disease: a systematic review of health utilities, resource use, costs, and economic evaluations of interventions. Value Health. 2019;22(11):1329–1344. doi:https://doi.org/10.1016/j.jval.2019.06.011

Response 9: Thank you for your valuable suggestion. We have added these citations to our latest manuscript (page 14, line 442, references 37-39).

Point 10: Authors should expand the conclusion section especially explaining what this study adds to the scientific literature. It is necessary that the authors investigate these aspects in depth, highlighting the areas in which these results could have healthcare implications.

Response 10: Thank you for your suggestions. We have modified the conclusion section by explaining what this study adds to the scientific literature and highlighting the areas in which these results could have healthcare implications (page 14, lines 479-499).

The modified conclusion is as follows:

“With the widespread and continuous mutation of SARS-CoV-2, PCV13 vaccination is not only a preventive strategy for pneumococcal diseases but also a preparation for the local epidemic of COVID-19 to protect vulnerable populations because of the high risk of co-infection [2, 3]. This study suggests that there is a substantial local demand for PCV13 vaccination among children under five years of age, but vaccine hesitancy still exists in this low-resource setting in China. Therefore, it is essential to promote PCV13 vaccination for children by ensuring an ample supply of vaccines and reducing parental concerns at the same time. People with vaccine hesitancy and refusal have different concerns, and targeted interventions should be applied. More publicity and education efforts on PCV13 may be an effective way to turn vaccine hesitancy into acceptance. New vaccination policies such as inclusion of PCVs in the NIP and publicly-funded PCV13 vaccination are needed to address vaccine refusal. In addition, a socially supportive environment, such as community engagement and positive social norm messages, may promote people to get vaccinated actively. The full payment for PCV13 vaccination is a financial burden for a significant number of people, but the partial payment was widely accepted, indicating that the local government’s financial support and reasonable pricing should be appreciated. This study helps understand the barriers to PCV13 vaccination in low-resource settings of China, and provides suggestions for local vaccination policy and intervention strategies. Findings in this study could also be valuable for other regions, especially other low-resource settings in China where the PCV vaccination coverage is rather low and the same barriers for vaccination may exist.”

Point 11: Discuss in depth potential strengths of the study within a specific paragraph.

Response 11: Thank you for your suggestion. We have added a specific paragraph to discuss potential strengths of the study (page 14, lines 452-467).

“This is the first study to investigate vaccine hesitancy regarding the administration of PCV13 in children and willingness to pay for it in low-resource settings in China. This study can help understand current status of vaccine hesitancy that may differ from previous studies, because people may have updated their knowledge and attitudes about vaccination after the outbreak of COVID-19. Three different status of vaccine hesitancy (i.e. vaccine acceptance, hesitancy, and refusal) are considered, and the reasons and associated factors for vaccine hesitancy and vaccine refusal are investigated. This can help deal with vaccine hesitancy more efficiently because targeted interventions can be provided for different people. To evaluate willingness to pay among caregivers, four possible payment schemes for vaccination in China are considered (i.e. full payment, partial payment, service charge only, and completely free), and the detailed willingness to pay is further investigated using the contingent valuation methods with a payment card approach. It will help understand economic affordability of residents in depth and provides an important reference for local vaccination policy makers when they set price for vaccines. Additionally, this study is a face-to-face interview survey, which could make our results more accurate and reliable.”

Round 2

Reviewer 3 Report

Thank the authors for their excellent work and great responsiveness (in time and quality).
The authors reply letter is excellent.
.The changes suggested were addressed positively.